# A Multidisciplinary Approach to Swallowing Rehabilitation in Patients with Forward Head Posture

**DOI:** 10.3390/medicina59091580

**Published:** 2023-08-31

**Authors:** Daiana Debucean, Judit Mihaiu, Adrian Marius Maghiar, Florin Marcu, Olivia Andreea Marcu

**Affiliations:** 1Doctoral School of Biomedical Sciences, University of Oradea, 410087 Oradea, Romania; daiana_debucean@yahoo.com; 2Faculty of Medicine and Pharmacy, University of Oradea, 410073 Oradea, Romania; amaghiar@gmail.com (A.M.M.); oli_baciu@yahoo.com (O.A.M.); 3Department of Psychoneuroscience and Rehabilitation, University of Oradea, 410087 Oradea, Romania; mfmihai27@yahoo.com

**Keywords:** atypical swallowing, forward head posture, rehabilitation, orofacial myofunctional therapy, manual therapy, Global Postural Re-education

## Abstract

(1) *Background and Objectives*: The forward head posture (FHP) is characterized by increased extensions of upper cervical vertebrae and flexion of the lower cervical vertebrae and upper thoracic regions, associated with muscle shortening. The compressive loading on the tissues in the cervical spine negatively impacts suprahyoid and infrahyoid muscles and generates increased tension of the masticatory muscles. The tongue has relations with the suprahyoid and the infrahyoid muscles. The pattern of swallowing evolves gradually from birth to the age of four. If this developmental transition does not occur, the result is persistent infantile or atypical swallowing—an orofacial myofunctional disorder with the tongue in improper position during swallowing, causing strain and stress on the jaw, face, head and neck. In FHP, muscles crucial to swallowing are biomechanically misaligned. The lengthening of the suprahyoid muscles necessitates stronger contractions to achieve proper hyolaryngeal movement during swallowing. This study assesses the added benefits of physiotherapy to the traditional myofunctional swallowing rehabilitation for patients with FHP. The underlying hypothesis is that without addressing FHP, swallowing rehabilitation remains challenged and potentially incomplete. (2) *Materials and Methods*: A total of 61 participants (12–26 years) meeting the inclusion criteria (FHP and atypical swallowing) were divided into two similar groups. Group A attended one orofacial myofunctional therapy (OMT) and one physiotherapy session per week, group B only one OMT session per week, for 20 weeks. Exclusion criteria were as follows: ankyloglossia, neurological impairment affecting tongue and swallowing, cervical osteoarticular pathology, other previous or ongoing treatments for FHP and atypical swallowing. (3) *Results*: There is a significant improvement in terms of movement and use of the orofacial structures (tongue, lips, cheeks), as well as in breathing and swallowing in both groups. Group A achieved better outcomes as the CVA angle was directly addressed by manual therapy and GPR techniques. (4) *Conclusions*: The combined therapy proved to be more effective than single OMT therapy.

## 1. Introduction

Human posture refers to the relationship between the body parts (head, neck, trunk, upper and lower limbs) in an upright position. It encompasses the position, the body shape, the dynamic and static balance and the neuromuscular mode of operation [1]. A healthy posture protects the supporting structures of the body against injury or progressive deformity, allowing the muscles to work most efficiently for effective movement and endurance [2].

There are three physiological curves that act to balance the human spine: cervical lordosis, thoracic kyphosis and lumbar lordosis, providing support and resistance against gravity forces. A proper posture maintains the body in a musculoskeletal balance with a minimal amount of stress and strain on the body, with the head in line with the ribs and hips and the back of the neck kept in line with the spine (in lateral view). The cervical spine allows the stability and the mobility of the head and neck; the upper cervical spine (C0–C2) is responsible for 50% of total neck flexion and extensions and for 50% of overall cervical rotation [3].

The modern lifestyle exposes people to dysfunctional postures [4], since people spend long periods on computers, mobile phones and game consoles or adopt faulty posture during reading. The forward head posture (FHP) is one of the commonly recognized types of poor head postures in the sagittal plane. It is clinically expressed by head tilt, uneven and rounded shoulders, spinal misalignment and curvature distortions. The FHP involves the following [5]:Increased extensions of the upper cervical vertebrae;Extension of the occiput on C1;Increased flexion of the lower cervical vertebrae and the upper thoracic regions.

This hyperextension of the upper cervical spine is associated with the shortening of the following muscles:Upper trapezius;Cervical extensors;Levator scapulae;Sternocleidomastoid.

Subsequently, mobility impairment is generated in the area. Sustaining the head in this forward posture for a prolonged period of time increases the compressive loading on the tissues in the cervical spine, leading eventually to muscular and skeletal disorders:The upper crossed syndrome—with reduction in lordosis in the lower cervical and kyphosis of the upper thoracic vertebra, shortening of the muscular fibers of the muscles involved in the atlanto-occipital articulation and overstretching of muscles around joint.Weakening of the respiratory muscles, with a negative impact on the respiratory function, the anterior throat muscles (suprahyoid and infrahyoid) and generating an increased tension of the masticatory muscles (the masseter, the pterygoid and the temporalis).

Studies acknowledged that a poorly positioned head, inclined backwards, forwards and sideways, alters the positioning of the tongue, depending on the deviation degree [6]. Tongue positioning impacts its function, so head posture and swallowing are interdependent variables [7]. Swallowing function is both directly and indirectly related to postures, such as head and cervical angle and body position [8,9,10,11].

Swallowing is a physiological life function of the body, implying a complex neuromuscular mechanism that determines the progression and transportation of the saliva, liquids and food bolus from the oral cavity to the stomach. It is induced by the impulses transmitted by the sensory receptors from the oral cavity, tongue and lips. In early childhood, the tongue is placed between the lips, permitting the suction effect; it is the “infantile swallowing” or “tongue thrust”, which is adequate for the age, the tongue performing a forward movement and putting pressure against the lingual surfaces of the anterior teeth. This swallowing pattern changes gradually into a mature, adult swallowing after the dental eruption, until the age of four. If it does not happen, infantile swallowing persists as atypical swallowing and is then considered a dysfunction [12,13].

The currently accepted model for swallowing describes three main stages: oral, pharyngeal, and esophageal phases.

The oral stage is often split into two sub-stages [14]:Oral preparatory—the solid food is processed through mastication and manipulation in the oral cavity and the liquids are sealed in the cavity by the tongue anteriorly and the hard palate posteriorly.Oral propulsion—the tongue elevation moves the bolus into the oropharynx.

During adult swallowing, the tip of the tongue touches the top front of the upper palate (the palatine spot) while the lower jaw makes a slight motion upward. This stage is under voluntary muscle control.

The pharyngeal phase begins when the bolus reaches the palatoglossal arch, and it is an irreversible step in the swallowing process. The bolus is directed into the esophagus, while the airways are protected by closing the nasopharynx (by elevation of the soft palate) and the vocal folds. Retroversion of epiglottis by the tongue helps direct the food bolus into the esophagus. The pharynx becomes elevated and pulled anteriorly by the contraction of the suprahyoid muscles to facilitate the pharyngeal–esophageal transition. The pharyngeal stage is much like a reflex.

The esophageal phase starts when the bolus is propagated inferiorly by a wave of peristalsis once it reaches the esophagus and ends once it passes into the stomach [15]. This is an autonomous process not under voluntary control, much like the pharyngeal phase. These phases work in a sequential and coordinated manner.

Weak contractions of the tongue and soft palate can cause premature leakage of the bolus into the pharynx, especially with liquids. Tongue dysfunction produces impaired mastication, bolus formation and bolus transport. These usually result from tongue weakness or incoordination. The oropharyngeal dysfunction can impair the swallowing initiation, an ineffective propulsion of the bolus or coughing and choking during swallowing due to the retention of a part of the bolus in the pharynx [14].

The relationship between posture and deglutition is bidirectional. The upward position of the tongue enhances the postural stability when standing on unstable surface without visual cues, so the tongue positioning modulates the postural control mechanisms [16].

Swallowing is made with ease when in upright position, but more difficult in head or neck flexion or extension [17,18], due to the decrease in airway or esophagus opening. In FHP, the muscles involved in swallowing are placed in an abnormal biomechanical position that may affect the ease of swallowing. Anatomically, the tongue has several relations with the hyoid bone, and therefore, with the suprahyoid and the infrahyoid muscles [19]. These muscles act together in the jaw and tongue movements, during the first phase of swallowing and phonation [20,21]. Electromyography showed electrical activity in the omohyoid muscle and in the anterior belly of the digastric muscle during different movements of the tongue; these muscles intervene to allow a proper association between the tongue and the head (neck), during flexion, extension and rotation of the neck and the cervical tract [20,22]. 

The suprahyoid muscular action helped in maintaining the posture and the equilibrium of the head. In FHP, the suprahyoid muscles adopt a kinematically disadvantageous position, their lengthening causing the need of more forceful contractions in order to attain adequate hyolaryngeal excursion during swallowing. The mandible is pulled into retrusion due to the stretch imposed on the suprahyoid muscles. The pharyngeal aponeuroses attach at the level of pharyngeal tubercle of the occipital bone and the tractions of these occur anteriorly to the C0–C1 joints. The proper functioning of the suprahyoid muscles is crucial in elevating the hyolaryngeal complex so this may explain the reason FHP negatively influences the swallowing mechanics when under this stress.

Abnormal/reverse/atypical swallowing is a myofunctional problem with the tongue in an improper position during the act of swallowing, which causes strain and stress on the jaw, face, head and neck. The tip of the tongue extends too far forward and down, the mandible moves backwards, which causes the head and neck to move in a forward motion to force the food back to the back of the throat. Typically, abnormal swallowing is associated with tongue thrusting, an open bite, a shallow palate and an underdevelopment of the bones in the upper jaw. Most patients have a long and narrow jaw and face and a head-forward posture. Other common signs include open lips and/or a tongue that sticks out when swallowing, protruding teeth, lisp, mouth breathing, neck and shoulder tension.

Considering the frequency of FHP among the general population [23,24], the aim of this research is to explore if a physiotherapy rehabilitation program targeting the FHP adds benefits to a myofunctional swallowing rehabilitation program within a healthy young population. Our rationale is that if FHP is not corrected, the rehabilitation process of the swallowing is difficult and sometimes incomplete, the results being endangered in the long term.

## 2. Materials and Methods

### 2.1. Experimental Group Characteristics

This study includes 61 young patients (36 females and 25 males) aged 12–26 years, with atypical swallowing and FHP, and took place between May 2021 and April 2023 at the specialized outpatient clinic of a private hospital in Oradea, Romania, at the physiotherapy and speech therapy departments.

This study was conducted according to the guidelines of the Declaration of Helsinki, and approved by the Commission for Research Ethics of the University of Oradea, Faculty of Medicine and Pharmacy, Romania (no. 22/18 November 2019). The purpose of this study and the precise methodology were explained to all subjects. Prior to their enrollment, all participants provided and signed informed consent, as well as the parents or legal representatives of all the minor subjects (aged under 18). They were also informed about the possibility to withdraw from this study at any time and with no consequences.

### 2.2. Inclusion Criteria

The general inclusion criteria were the atypical swallowing (previously diagnosed by general dentists or orthodontists) and the forward head posture (diagnosed by a general practitioner—family doctor, a neurologist or a physiotherapist).

### 2.3. Exclusion Criteria

The exclusion criteria were ankyloglossia, neurological impairment affecting the tongue and swallowing, cervical osteoarticular pathology, other previous or ongoing treatments for FHP and atypical swallowing.

### 2.4. Research Plan

The conducted trial was an analytic observational cohort study [25].

Subsequent to the initial myofunctional and physiotherapy assessment, the subjects were informed about the results and the two rehabilitation program possibilities: either Orofacial Myofunctional Therapy (OMT) alone or OMT combined with physiotherapy, consisting of manual therapy (MT) and Global Postural Re-education (GPR). Two groups were formed, based on the choice and consent expressed by each subject: group A (combined therapy) and group B (OMT alone). Initially, there were 69 subjects (39 females and 30 males), but 8 of them (3 females and 5 males) gave up after variable times and did not complete the rehabilitation program. The dropouts occurred due to personal reasons (5—difficulties in synchronizing the professional schedule with the therapies; 3—illness). The 61 subjects completed the program (Figure 1).

The time of therapy for each patient covered a period of 20 weeks, with one weekly 50 minutes’ session of physiotherapy (manual therapy + GPR) and one weekly 40 minutes’ session of OMT, always scheduled as follows: first, the physiotherapy session, then, within 2 days, the OMT session. The patients also received personalized at-home exercise plans both from the physiotherapist (lasting for 30 min) and the myofunctional therapist (lasting for 15 min) for daily training.

### 2.5. Assessment

Two assessments were performed for each patient in each specialty: an initial assessment, before the beginning of the therapies, and a final one, after 20 weeks, at the end of the rehabilitation programs. The assessments consisted of a myofunctional assessment (in 2 sessions) performed by a speech therapist specialized in OMT and a postural assessment in 1 session, performed by a skilled physiotherapist.

#### 2.5.1. The Myofunctional Assessment

Each patient was referred for a myofunctional assessment by a general dentist or an orthodontist (before the start of the orthodontic treatment) with the suspicion of an orofacial myofunctional disorder. The assessment comprises an extensive anamnesis, with questions related to the following:Birth and developmental history, speech, language and hearing history;Interventions such as speech therapy, physical therapy, dental and orthodontic interventions or devices used;Medical history (conditions that may affect oral functioning, such as repeated upper airway and ear infections, respiratory allergies, snoring, head and neck injuries), surgery history (frenectomy, tonsillectomy, adenoidectomy, orthognathic surgery);Breathing (nasal or mouth breathing);Oral habits (non-nutritive sucking habits—thumb, digit, object sucking);Feeding history (avoiding food textures that need increased oral manipulation and chewing, tendency to drink liquids to help swallowing, chewing with mouth open, noisy chewing and swallowing, excessive slow eating, muscle fatigue when chewing).

The Orofacial Myofunctional Evaluation with Scores-Expanded (OMES-E) [26] was used to assess the orofacial structures and functions. The mean coefficients are as follows: sensitivity 0.91, specificity 0.77, positive predictive value 0.87, negative predictive value 0.85.

The visual examination for appearance and posture evaluated the following orofacial structures:Face—symmetry between right and left sides, proportion between the facial thirds, the nasolabial sulcus;Cheeks—volume, configuration, tension;Mandible/maxilla relation—free space, midline, malocclusion or malposition;Lips—volume, configuration, posture at rest, commissures;Mentalis muscle—contraction at rest;Tongue—volume, position at rest;Palate—height, width.

Scores were attributed on a 4-point scale: 4—normal, 3—mild alteration, 2—moderate alteration, 1—severe alteration.

Additionally, we examined the profile, the lingual and labial frenulum, the volume of the tonsillar tissue, the soft palate configuration, the sensitivity inside and outside the mouth. The oral rest posture is then checked carefully; a typical, healthy rest posture is with the lips closed, teeth slightly apart (1–2 mm physiological interdental distance), with the tip of the tongue resting against the anterior hard palate and nasal breathing.

The next step in assessment was the movement symmetry and mobility of the lips (protrusion, retrusion, lateral to right, lateral to left), tongue (protrusion, retrusion, lateral to right, lateral to left, raising, lowering), cheek (inflate, suck, retract, transfer air from one part to the other), jaws (opening, closing, right laterality, left laterality, protrusion) and velum. Lack of precision in the movement, tremor, associated movements of other components (lips or jaw accompanying the movements of the tongue) and/or the inability to perform the movement were considered dysfunction. The examiner attributed scores on a 6-point scale: 6—normal, 5—insufficient ability, 4—insufficient ability and associated movements, 3—insufficient ability and tremors and/or deviation, 2—insufficient ability, associated movement tremors and/or deviation, 1—absence of ability to perform the movement.

The functional assessment analyzed the breathing mode, the deglutition and the mastication.

Breathing was observed throughout the evaluation and was classified as follows:Nasal—the lips remained in occlusion without effort, mainly during situations of rest and mastication, with the tongue contained in the oral cavity—4 points;Mouth breathing—mild alteration—3 points (mouth breathing, but the ability of nose inspiration preserved, without showing signs of fatigue and dyspnoea); moderate alteration—2 points (similar to the previous one, but the nasal pattern could not be maintained); severe alteration—1 point (while trying to perform nasal only inspiration, signs of fatigue and dyspnoea appeared and the subject opened his mouth to inspire within a few seconds, both at rest and during mastication).

The deglutition analysis comprises the following:
Lip behavior: closure without effort—6 points; closure with effort or with tongue between dental arches—4/3/2 points, according to the degree of dysfunction; absence of lip closure—1 point;Tongue behavior: in the oral cavity during swallowing—4 points; between the dental arches or interposed with the teeth (when an overbite or overjet is present)—3/2/1 point(s), depending on the severity of the dysfunction and noticing the place of the interposition;Other behaviors and change signs: head or body parts movements, mandible sliding, facial muscle tension, food leakage, choking or noise; presence—1 point, absence—2 points;The efficiency of the deglutition—separately for food and for liquids: one deglutition (the bolus passes from the oral cavity into the oropharynx with a single movement)—3 points, 2–3 repetitions needed (2–3 attempts to perform a complete deglutition)—2 points, multiple repetitions to succeed (multiple attempts to achieve one deglutition)—1 point.

The assessment of the mastication aimed at the following:The type of mastication: bilateral—alternated 10 points, simultaneous (vertical) 8 points; unilateral—grade 1 (61–77%) 6 points, grade 2 (78–94%) 4 points, chronic (95–100%) 2 points; anterior 2 points; inability to masticate 1 point;The bite: normal (incisors) 4 points; canines-premolars 3 points; molars 2 points; inability to bite 1 point;Other behaviors and change signs associated with mastication: movements or altered posture of the head or body parts, food leakage—presence 1 point, absence 2 points.

Another functional assessment of the swallowing was performed using a traditional protocol, the Payne technique [27]. The subjects were asked to stick their tongues out. The excess of saliva was cleaned and the fluorescent Payne pasta was applied with a spatula on the right edge, the frontal area and the left edge of the tongue. Subjects were asked to swallow only once and then open their mouth. A Payne lamp was used to visualize the impressions of the fluorescent Payne pasta. The traces of the Payne pasta on the palate and tongue indicate the contact points of the tongue and its movements during the oral phase of swallowing and whether there is an abnormal swallow. The swallowing traces can be seen.

The assessment of the lingual frenulum was necessary because a restrictive frenulum interferes with the rehabilitation of swallowing, and we considered it a temporary exclusion criterion until frenotomy is conducted—if the assessment shows the need for surgery. We used the Hazelbaker Assessment Tool for Lingual Frenulum Function (HATLFF) [28]. It comprises appearance items (lifted tongue, frenulum elasticity, length of frenulum when tongue is lifted, place of attachment to the tongue and to the inferior alveolar ridge) and function items (tongue lateralization, lifting, extension, spread of anterior tongue, cupping, peristalsis, snapback). A score of 14 is perfect, 11 is acceptable if appearance score is 10, and a score less than 11 shows impaired function and need for case management. If the appearance score is less than 8, frenotomy is necessary to establish the proper function.

The last step of the myofunctional assessment was the evaluation of specific speech sounds production:The placement of tongue tip for /t/,/d/,/n/ and /l/;The position of the tongue when articulating /s/, /z/, /ʃ/, /tʃ/, /ɜ/, /dɜ/—erroneously produced interdentally, with lateralization or obviously against the anterior dentition;Distortion of the velar sounds /k/ and /g/;Distortion of or inability to produce the /r/ sound;Any deviations of the jaw during connected speech;Nasalization of the vowels;Weak bilabial productions;Diadochokinetic tasks—slower rates in diadochokinetic tasks were associated with postural differences [29]: on single-syllable /pΛ/ measure, slower rates were associated with open-mouth postures; on trisyllabic /pΛtΛkΛ/ measure, slower rates were correlated with dentalized postures of the tongue.

#### 2.5.2. The Postural Assessment

There is no standard procedure for posture analysis. The assessment of the FHP is conducted through few methods:-The radiography—X-ray scan—although it offers a clear image of the reference points and it is considered the golden standard, it involves radiation and higher costs, so it is neither practical nor chosen in studies.-The observational analysis of the head position with reference to some anatomical landmarks, in a side view—an imagined vertical-line passes through certain external anatomical landmarks: ear lobe (mastoid process)—acromion-clavicular joint—hip—knee—foot [26].-The craniovertebral angle (CVA)—it is among the most reliable and common methods for assessing FHP [30]. It is the intersection of two lines: a horizontal line passing through the C7 spinous process and a line joining the midpoint of the tragus of the ear to the skin overlying the C7 spinous process [31]—see Figure 2.

Participants were informed before the evaluation. We marked the tragus and the spinous process of C7 with a body marker and took 2 lateral photos of each subject in standing position—it is a more sensitive posture to evaluate the FHP [32]. The 12 megapixels camera used for taking photographs was placed 1.5 m away from the subjects on a tripod, at a height of 115 cm. The subjects were invited to stand barefoot in a comfortable position, with both arms relaxed at the sides of the trunk and to maintain the natural head posture, on a spot on the ground, previously marked, and to look ahead. For the exact measurement of CVA, a plumb line hung from the ceiling descended right next to the subject to draw a horizontal line [33]. The clinical use of the photographic posture analysis is recommended in the literature because it is an accurate and objective method [34].

The angle formed of the horizontal and the vertical lines was measured with MB ruler (Markus Bader—MB Software Solutions, triangular screen ruler) and the result was recorded. Cronbach’s α coefficient value of intra-rater (0.999) and inter-rater (0.892) reliabilities are high; thus, MB ruler software is reliable for assessing the CVA [35,36]. A CVA less than 48–50 defines the FHP. The smaller the CVA, the greater the FHP. The CVA measurement has a good reliability and validity in the measurement of FHP [37]. Data were collected and input on a excel spread sheet and then analyzed.

### 2.6. Study Endpoints

The study endpoint is the effect of including the MT and GPR in the tongue posture and swallowing pattern rehabilitation compared with OMT alone.

### 2.7. Rehabilitation Program

#### 2.7.1. OMT Rehabilitation Program

The OMT is a neuromuscular re-education of the muscle functioning to correct the tongue, lips and cheeks’ rest posture and movements, as well as the breathing, mastication and swallowing patterns. It includes exercises designed to improve the proprioception, tone and mobility. For our subjects, the aim of the OMT intervention is to make them aware of the incorrectness of the tongue position and functioning and to harmonize the orofacial functions by learning a new physiological swallowing pattern. A correct myofunctional protocol is adapted to the needs of each person and its success depends to a great extent on the compliance of the subjects (in the case of children, both the kids and the parents) [38,39].

The step-by-step goals are as follows:Nasal breathing—although not all of our subjects are mouth breathing, this is a common issue in atypical swallowing. Each of them completed breathing exercises, but more emphasis was placed in case of those with oral breathing. Based on neuroplasticity, the exercises aim to re-train the brain to use a new physiological routine of breathing. Nasal breathing and the lips sealed facilitate the tongue to adopt an upward position.Training the tongue with a new rest position—the exercises stimulate the anterior tongue, then the lateral parts and the posterior tongue. The final goal of this step is teaching the tongue to rest with the tip on the retroincisal papilla and the remaining part on the palate. A tongue resting on the hard palate helps the restoration of the lip seal and nasal breathing. The exercises target the oral tactile stimulation, tongue tip elevation and stability, tongue movements without involving the mandible, and the lingual-palatal seal, the development of the midline groove necessary for the bolus control.Restoring the lip seal—the atypical swallowing is performed with an excessive contraction of the orbicularis oris muscle to compensate the lip incompetence; the lip exercises (closure and competency exercises) will restore the physiological lip seal at rest.Increasing the facial muscle tone—improving the buccinator and masseter muscles functioning will result in better chewing, correct swallowing and mandible stability.Restoring the tone of the soft palate—exercises of raising and lowering the soft palate were performed.Re-patterning the correct swallowing—it is addressed when the physiological breathing and chewing are restored, the correct resting position of the tongue and a good lip seal are achieved. The swallowing exercises gradually use thin liquids, thick liquids, solid food.

The OMT Rehabilitation Program consisted of 20 weekly sessions of 40 min each, with a personalized set of exercises performed every week. At the end of every session, each subject received a set of 3–4 exercises to perform at home daily, in front of a mirror, until the next session.

#### 2.7.2. MT and GPR Rehabilitation Program

The goals in the physiotherapy management of the FHP are as follows:The postural alignment and balance (using cervical and scapular retraction exercises);The increase in the joint mobility and flexibility (by cervical traction, thoracic manual techniques and exercises, stretching exercises of the trapezius, scalene, pectoralis muscles and sternocleidomastoid);The reduction in the muscular spasm (by myofascial release, position release techniques);Muscle strengthening and endurance (isometric strengthening exercises for the cervical region, gradually changing to isotonic and then dynamic exercises).

All these exercises are delivered through two major approaches: the MT and the GPR.

MT is a conservative treatment provided by physical therapists, as an effective modulation in relieving soft tissue [40,41,42]. It was performed for 20 min at the beginning of each session and included a combination of therapeutic techniques, personalized for the need of each patient:Long axis traction / distraction of C0–C2;Mobilization of the cervical spine;Mobilization of the upper thoracic spine;Release of the infra-mandibular, sternocleidomastoid and suprahyoid muscles with the stabilization of the hyoid;C0–C1 joint mobilization in extension for the flexor muscles (especially the deep small intrinsic periarticular muscles);Suboccipital muscle inhibition;Manual passive stretching on the pectoralis minor, scalene, upper trapezius, shoulder retractors, deep cervical flexors and cervical extensors;Therapeutic massage of the neck and shoulders.

FHP, with posterior cranial rotation and stretching of the infrahyoid muscles, leads to an increased activity of the masticatory muscles and cranial extensors [7,8,9,10]. The muscles of mastication will try to maintain the mandible up (mouth closed, lips touching) and the infrahyoid muscles are trying to bring the mandible down and back; the result is a continuous fight between the mandible depressing and elevating muscles. We must balance these muscles to restore the normal alignment of the craniovertebral angle.

The at-home exercises were designed for posture improvement and stretching. The techniques and exercises were only partly the same for all the patients; the plans were personalized according to the complaints of each patient and the assessment performed. the patients with oral breathing performed more breathing and lip exercises to achieve nasal breathing pattern and proper lip seal.

GPR is a method substantiated by Souchard, a French physical therapist, in the 1980s and nowadays used worldwide [43,44,45] in treating patients with musculoskeletal disorders and impairments, including those of the cranio-cervico-mandibular system [46,47,48,49]. It is based on the global stretching of the antigravitational muscles organized in muscle kinetic chains [50], being clinically proven to reduce postural impairments and regain muscle symmetry and adequate posture through global active muscular stretching postures and joint decompressions [51]. Its techniques and exercises involve motor control, contractions of the antagonist muscles and sensory integration in order to improve mobility, flexibility, muscle strength and functioning for muscle balance and postural symmetry.

When planning the postural corrective exercise series, we took into consideration the shortening or lengthening of the muscles, as well as their hypo- or hyperactivity, and the position in which each patient spends most of the time during his/her daily activity (standing or sitting at work or during other daily activities). Our GPR sessions lasted 30 min and consisted of a series of specific muscle chain stretching positions, evolving gradually from a minimum tension to a greater one, through progressive stretching:Stretching of the anterior muscular chain;Stretching of the posterior muscular chain—bending forward position;Active global stretching.

At the end of each session, the subjects were requested to correct their standing posture (including the entire spine and the pelvis).

Each patient had to do some at-home daily exercises designed to improve the therapeutic effects of the sessions in the physiotherapist’s office. They were stabilizing and posture correction stretching exercises: chin tuck in sitting; upper truck extension with chin tuck; stretching exercises targeting the stabilization of the periscapular muscles; self-stretching exercise for pectoralis minor; sternocleidomastoid and levator scapulae stretching; scalene and upper trapezius stretching in prone; stretching exercises for the shoulder retractors and deep cervical flexors, for pectoralis muscles and cervical extensors.

There are studies stating that individuals with FHP who received a combination of upper thoracic spine mobilization and mobility exercises demonstrated better overall short-term outcomes in terms of the CVA in standing position [52,53].

The entire home training lasted 30 min.

The patients were also taught to adopt ergonomic measures, consisting of ergonomics at work (special chair at the office) and ergonomics when sleeping—sleeping on the back with a smaller pillow so as not to induce a greater flexion.

### 2.8. Outcome Variables

In the first pre-intervention assessment, all variables were measured, including the sociodemographic variables (age, sex). Later, all outcome variables were measured in the final assessment after completing the agreed rehabilitation program. The variables (except the sociodemographic ones), which are also outcome measures, and the data collection tools are listed in Table 1. 

### 2.9. Statistical Analysis

Statistical analysis was performed on the statistical package SPSS 24 (Armonk, NY, USA: IBM). Patients were under observation in two different moments, pre- and post-rehabilitation. To test the differences between the variables as measured before and after the rehabilitation program, we used the paired *t*-test in both groups.

Two tails paired *t*-tests are used when each unit is measured twice, resulting in a pair of observed data. The test is applied to assess if the mean difference is zero.

When comparing the results of group A to group B, because the observation is not made on the same patient, but on independent samples, the differences are assessed using independent *t*-test. This is applied both for the pre- and the post-rehabilitation moments, in two separate analyses.

For all the analysis, it was considered a significance level of 0.05.

## 3. Results

A total of 61 young patients aged 12–26 years were involved in our study, as follows: 36 women aged 12–26 years and 25 men aged 14–24 years. The patients’ characteristics in each group are presented in Table 2. Out of the total number of patients, 59% were women and 41% men. The analyzed variables, which are also the outcome measures, were the items of the OMES-E (AP, ML, MoT, MJ, MC, Br, DLB, DTB, DOB, DE, TDR, MB, MaT, MOB, TMR) and the CVA.

Group A is composed of 12 (38.7%) men and 19 (61.3%) women, while group B is composed of 13 (43.4%) men and 17 (56.6%) women. There is a significant difference in sex distribution in the two groups. The average age in each group is 19 years, with no significant difference in patient age distribution in the two groups.

Table 3 presents the means and standard deviations for both groups and both treatment strategies and the paired *t*-test outcomes. Testing is assessed separately for group A and also for group B, but all the results are summarized in one table.

There are significant differences for all variables at the moment of the final assessment, after the program completion, compared to the initial assessment, both for A and B groups, except for the MOB variable in B group.

It can be noticed in group A that the mean values after treatment are greater than in before treatment, progress also confirmed by significant paired *t*-test results: AP (18.93 to 19.87, *p*-value < 0.001), ML (17.54 to 20.52, *p*-value < 0.001), MoT (23.54 to 31.03, *p*-value < 0.001), MJ (24.32 to 26.51, *p*-value < 0.001), MC (19.9 to 21.12, *p*-value < 0.001), BR (3 to 3.67, *p*-value < 0.001), DLB (4.16 to 5.32, *p*-value < 0.001), DTB (1.93 to 3.61, *p*-value < 0.001), DOB (9.06 to 11.45, *p*-value < 0.001), DE (4.8 TO 5.9, *p*-value < 0.001), TDR (19.96 to 26.29, *p*-value < 0.001), MB (3.54 to 3.83, *p*-value < 0.001), MaT (4.19 to 7.87, *p*-value < 0.001), MOB (5.58 to 5.93, *p*-value < 0.001) and TMR (13.35 to 17.64, *p*-value < 0.001). For all variables, the results improved significantly after therapy.

In group B, there are significant improvements after therapy for most of the variables, AP (18.73 to 20.56, *p*-value < 0.001), ML (18.66 to 20.9, *p*-value < 0.001), MoT (24.43 to 30.6, *p*-value < 0.001), MJ (23.93 to 24.83, *p*-value < 0.001), MC (20.26 to 20.93, *p*-value < 0.001), BR (3.16 to 3.76, *p*-value < 0.001), DLB (4.13 to 4.96, *p*-value < 0.001), DTB (1.8 to 3.56, *p*-value < 0.001), DOB (9.1 to 10.8, *p*-value < 0.001), DE (4.96 to 5.26, *p*-value = 0.005), TDR (20 to 24.6, *p*-value < 0.001), MB (3.4 to 3.63, *p*-value = 0.006), MaT (5.26 to 6.53, *p*-value < 0.001), TMR (14.4 to 16.03, *p*-value < 0.001), except for MOB variable, where the results improved from 5.76 to 5.86 but not statistically significant (*p*-value = 0.184).

The CVA outcome measure change between the initial and the final assessment for both groups is presented in Table 4. In group A, there was an increase from 42.4 to 49.05, statistically significant (*p*-value < 0.001) while in group B the average increased significantly from 41.29 to 42.29 (*p*-value < 0.001)

Table 5 shows the Student’s *t*-test results at pre- and post-intervention assessments for group A compared to group B.

At the initial assessment, there are no significant differences among the characteristics of the patients in group A and group B: AP (18.93-A, 18.73-B, *p*-value = 0.828), ML (17.54-A, 18.66-B, *p*-value = 0.92), MoT (23,54-A, 24.43-B, *p*-value = 0.221), MJ (24.32-A, 23.93-B, *p*-value = 0.618), MC (19.9-A, 20.26-B, *p*-value = 0.401), BR (3-A, 3.16-B, *p*-value = 0.265), DLB (4.16-A, 4.13-B, *p*-value = 0.718), DTB (1.93-A, 1.8-B, *p*-value = 0.361), DOB (9.06-A, 9.1-B, *p*-value = 0.977), DE (4.8-A, 4.96-B, *p*-value = 0.389), TDR (19.96-A, 20-B, *p*-value = 0.885), MB (3.54- A, 3.4-B, *p*-value = 0.483), MaT (4.19-A, 5.26- B, *p*-value = 0.17), MOB (5.58-A, 5.76-B, *p*-value = 0.17), TMR (13.35-A, 14.4-B, *p*-value = 0.244).

After completing the agreed rehabilitation program, there can be noticed significant differences in MJ (26.51-A, 24.83-B, *p*-value < 0.001), DOB (11.45-A, 10.8-B, *p*-value = 0.005), DE (5.9-A, 5.26-B, *p*-value < 0.001), TDR (26.29-A, 24.6-B, *p*-value = 0.001), MaT (7.87-A, 6.53-B, *p*-value = 0.008) and TMR (17.64-A, 16.03-B, *p*-value = 0.003), all variables achieving a higher score in group A. For all the other variables, the changes are not significantly different: AP (19.87-A, 20.56-B, *p*-value = 0.087), ML (20.52-A, 20.9-B, *p*-value = 0.451), MoT (31.03-A, 30.6-B, *p*-value = 0.399), BR (3.67-A, 3.76-B, *p*-value = 0.445) have average scores higher in group B while MC (21.12-A, 20.93-B, *p*-value = 0.521), DLB (5.32-A, 4.96-B, *p*-value = 0.158), DTB (3.61-A, 3.56-B, *p*-value = 0.719), MB (3.83-A, 3.63-B, *p*-value = 0.119) and MOB (5.93-A, 5.86-B, *p*-value = 0.378) have higher average scores in group A.

As presented in Table 6, CVA has assessed significant differences in the pre- (42.4-A, 41.29-B, *p*-value = 0.02) and post-intervention (49.05-A, 42.29-B, *p*-value < 09.001) moments, with higher average scores in group A.

## 4. Discussion

There were significant differences assessed in the final evaluation, after 20 weeks of therapy, in every variable, in each of the two groups: group A (OMT + manual therapy and GPR) and group B (OMT). It means that the OMT reached its goals, meaning to modify the functioning of the targeted oral structures.

Generally speaking, all variables reached higher scores in group A, in which the rehabilitation program meant both OMT and physiotherapy, compared to group B where the intervention was only OMT.

The results were comparatively higher in group A than group B in the mobility of jaws and cheeks, deglutition lip behavior, deglutition other behaviors, deglutition efficiency, mastication bite, mastication other behaviors. The scores for appearance were higher in group B.

In both groups, the most notable differences (measured in score differences) were achieved in the following (see Table 3):Mobility of the tongue—from a mean of 23.54 to 31.03 (31.81%) in group A and 24.43 to 30.06 (23.04%) in group B;Deglutition tongue behavior—from 1.93 to 3.61 (87.04%) in group A and 1.8 to 3.56 (97.77%) in group B;Total deglutition result—in group A from 19.96 to 26.29 (31.71%), in group B from 20 to 24.6 (23%).

In group A, the greater differences pre- and post-rehabilitation were measured in mastication type and total mastication result (as seen in Table 3).

Jaw movement was impaired especially in laterality movements and protrusion, and much of this impairment was rehabilitated with the patients in group A, due to the combined intervention which addresses the involved muscles directly (the manual therapy and GPR) and indirectly (the OMT) [54].

We know that the tongue, lips and jaw function together as a unit [22] from birth until the age of 6–9 months, when they start separating their movements. This will allow jaw stabilization, the tongue consequently learning to pull itself from the jaw to achieve a proper resting posture, proper mastication movements and sound production clarity. Many of our patients had difficulty in disassociating the tongue from the mandible and this led to imprecise speech and this dissociation was one of our early goals in the OMT approach [55]. Some of them passed the diadochokinetic tasks compensating with the mandible rather than using the tongue. Independent use of each part is necessary for the refined movements required for a mature pattern of deglutition and speech. Correct positioning and movement of the tongue and lips are based upon stability of the jaw (referred to as external stability) and the tongue’s ability to remain in a neutral position in relation to the jaw (referred to as internal stability) [56].

The imprecise articulation of some sounds may be related not only to the inability to separate the mandibular and lingual excursions within the oral cavity, but also to the incorrect resting posture of the tongue and mandible, because this is the place from where the speech production begins and ends [57]. If the resting pattern remains the same, the traditional speech therapy will not be successful in correcting the altered sounds.

The change of the breathing mode was also an important goal, since mouth breathing is correlated with problems in swallowing and chewing [58,59]. Studies highlight that mouth breathing decreases chewing activity and reduces the vertical effect upon the posterior teeth [60,61]. Also, the permanent low posture of the tongue in mouth breathers leads to atypical swallowing with tongue pressure over anterior teeth and not against the palate, with the tip of the tongue on the incisive papilla [62].

The tongue movement improvement was seen in the manner of handling and swallowing the liquids and foods. From a forward or interdental protrusion of the tongue tip, the pattern progressed to the mature pattern with the tongue pressing up to the palate, with very little or absent face, cheek, lip or neck movement.

Manual therapy techniques used in the cervicothoracic area and GPR in our FHP patients improved not only the CVA, but the overall results of the OMT. Correcting the FHP was not addressed in the OMT, but occurred somehow “naturally” and in a very low percentage; the change was dramatic and obvious in group A (from a mean of 42.4 to 49.05, a normal angle), as well as in all the outcome variables, proving that the MT and GPR helped in the rehabilitation of the swallowing. These changes may involve the suboccipital muscles, as they act to maintain the posture of the head, in synergy with global chains and in relation to input from the visual, mandibular and swallowing systems [7,8,63,64].

We must consider some of the limitations of our study. First, the range of the age of our subjects is 12–26 years (mean age 19.15). Extending this study to smaller children or older adults may provide important clues about the efficiency of the therapy combination in correlation with age groups. Secondly, the small sample size (61 subjects) together with small age range are not an accurate representation of the general population with this condition (atypical swallowing and FHP). Thirdly, even if there are clinical tests used to diagnose the type of swallowing (normal or atypical), the experience of the examiner is very important in evaluating the involvement of the muscles and other compensatory components in swallowing.

## 5. Conclusions

Our study results demonstrate the efficacy of a multidisciplinary approach to the swallowing rehabilitation in young patients with FHP instead of the traditional single approach: although it is essential to set up a myofunctional rehabilitation procedure to correct the oral habit, granting long-term permanent results requires the involvement of a physiotherapist if there are postural issues. The connections of the muscles in the cranio-cervico-mandibular complex should be considered to optimize the clinical examination of the tongue and therefore enhance rehabilitation programs and therapeutic results.

There is a lack of evidence in the literature defining the ideal age to begin OMT and few studies assessing the head posture in children with atypical swallowing [11]. The knowledge of this fact may be useful in the diagnosis of atypical swallowing and its treatment, because in this case the rehabilitation should not be limited only to the orofacial structures but include the rehabilitation of the FHP. 

## Figures and Tables

**Figure 1 medicina-59-01580-f001:**
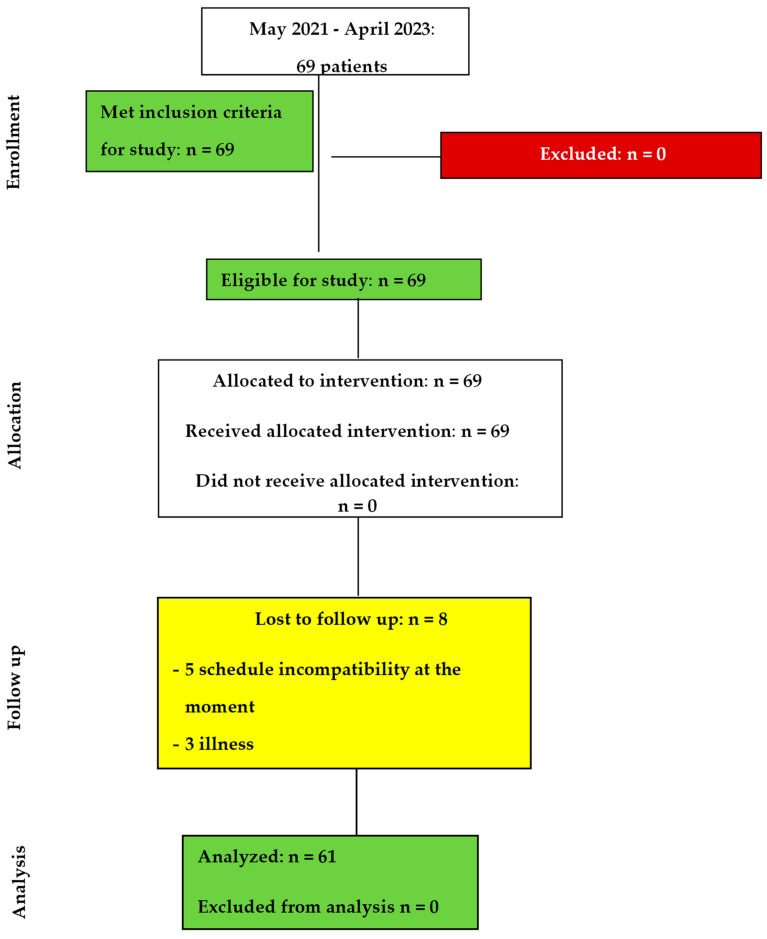
The flowchart of this study.

**Figure 2 medicina-59-01580-f002:**
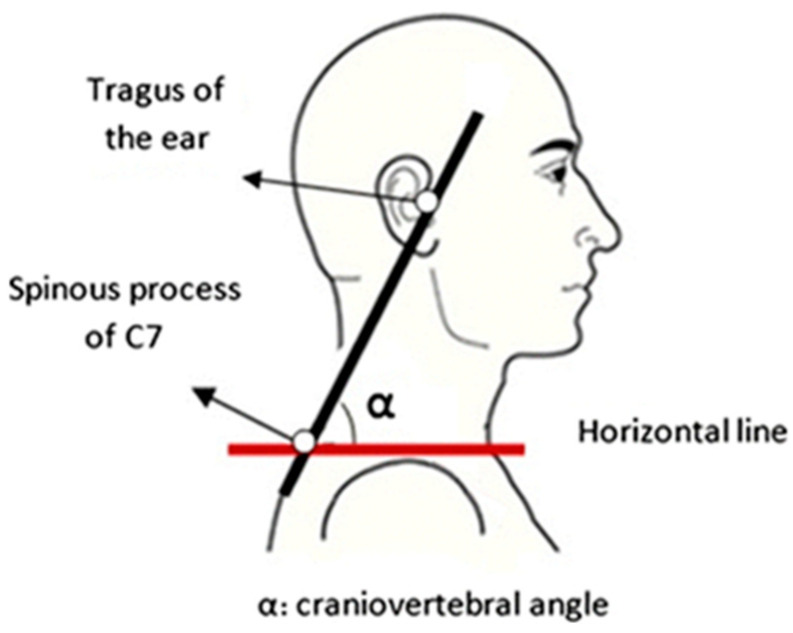
The craniovertebral angle [32].

**Table 1 medicina-59-01580-t001:** Summary of outcome variables.

Outcome Measure	Data Collection Tools
AP	OMES-E
ML	OMES-E
MoT	OMES-E
MJ	OMES-E
MC	OMES-E
Br	OMES-E
DLB	OMES-E
DTB	OMES-E Payne technique
DOB	OMES-E
DE	OMES-E
TDR	OMES-E
MB	OMES-E
MaT	OMES-E
MOB	OMES-E
TMR	OMES-E
CVA	2 lateral photos MB Ruler

AP—appearance and posture; ML—mobility of the lips; MoT—mobility of the tongue; MJ—mobility of the jaws; MC—mobility of the cheek; Br—breathing mode; DLB—deglutition lips behavior; DTB—deglutition tongue behavior; DOB—deglutition other behaviors; DE—deglutition efficiency; TDR—total deglutition result; MB—mastication bite; MaT—mastication type; MOB—mastication other behaviors; TMR—total mastication result; CVA—craniovertebral angle; OMES-E—Orofacial Myofunctional Evaluation with Scores-Expanded.

**Table 2 medicina-59-01580-t002:** Descriptive statistics.

Variable	Group A	Group B	*p*-Value	Result (Significant/ Not Significant)
Sex n (%)				
M	12 (38.7%)	13 (43.4%)	<0.001	Significant
W	19 (61.3%)	17 (56.6%)		
Age mean ± std	19.32 ± 3.58	19.03 ± 3.01	0.736	Not significant

**Table 3 medicina-59-01580-t003:** The comparison between the initial assessment (Pre) and the final assessment (Post) variables for group A and B.

Outcome Measure/Group	Pre Mean ± Std	Post Mean ± Std	T Value	*p* Value	Result (Significant/Not Significant)
AP (A)	18.93 ± 2.0	19.87 ± 1.83	−4.518	<0.001	Significant
AP (B)	18.73 ± 1.92	20.56 ± 1.22	−5.894	<0.001	Significant
ML (A)	17.54 ± 2.82	20.52 ± 2.03	−8.803	<0.001	Significant
ML (B)	18.66 ± 1.68	20.9 ± 1.56	−8.55	<0.001	Significant
MoT (A)	23.54 ± 2.41	31.03 ± 1.66	−21.033	<0.001	Significant
MoT (B)	24.43 ± 2.19	30.6 ± 2.25	−7.694	<0.001	Significant
MJ (A)	24.32 ± 2.57	26.51 ± 1.6	−6.412	<0.001	Significant
MJ (B)	23.93 ± 2.36	24.83 ± 1.91	−4.506	<0.001	Significant
MC (A)	19.9 ± 1.85	21.12 ± 1.35	−4.85	<0.001	Significant
MC (B)	20.26 ± 1.22	20.93 ± 0.98	−3.44	<0.001	Significant
Br (A)	3 ± 0.77	3.67 ± 0.47	−5.78	<0.001	Significant
Br (B)	3.16 ± 0.69	3.76 ± 0.43	−5.288	<0.001	Significant
DLB (A)	4.16 ± 1	5.32 ± 0.87	−5.887	<0.001	Significant
DLB (B)	4.13 ± 0.93	4.96 ± 1.06	−4.631	<0.001	Significant
DTB (A)	1.93 ± 0.67	3.61 ± 0.49	−15.584	<0.001	Significant
DTB (B)	1.8 ± 0.55	3.56 ± 0.5	−19.199	<0.001	Significant
DOB (A)	9.06 ± 0.96	11.45 ± 0.67	−12.605	<0.001	Significant
DOB (B)	9.1 ± 0.84	10.8 ± 1.03	−7.71	<0.001	Significant
DE (A)	4.8 ± 0.7	5.9 ± 0.3	−9.382	<0.001	Significant
DE (B)	4.96 ± 0.71	5.26 ± 0.58	−3.071	0.005	Significant
TDR (A)	19.96 ± 2.24	26.29 ± 1.41	−15.808	<0.001	Significant
TDR (B)	20 ± 1.81	24.6 ± 2.22	−12.324	<0.001	Significant
MB (A)	3.54 ± 0.8	3.83 ± 0.37	−2.334	0.026	Significant
MB (B)	3.4 ± 0.85	3.63 ± 0.61	−2.971	0.006	Significant
MaT (A)	4.19 ± 2.02	7.87 ± 1.45	−11.903	<0.001	Significant
MaT (B)	5.26 ± 2.85	6.53 ± 2.28	−4.08	<0.001	Significant
MOB (A)	5.58 ± 0.62	5.93 ± 0.24	−3.558	0.001	Significant
MOB (B)	5.76 ± 0.5	5.86 ± 0.345	−1.361	0.184	Not significant
TMR (A)	13.35 ± 2.3	17.64 ± 1.56	−12.025	<0.001	Significant
TMR (B)	14.4 ± 3.15	16.03 ± 2.38	−4.997	<0.001	Significant

**Table 4 medicina-59-01580-t004:** CVA variable in group A and B, pre- and post-intervention.

Outcome Measure/Group	Pre Mean ± Std	Post Mean ± Std	T Value	*p* Value	Result (Significant/Not Significant)
CVA (A)	42.4 ± 1.83	49.05 ± 1.56	−20.259	<0.001	Significant
CVA (B)	41.29 ± 1.89	42.29 ± 1.95	−8.808	<0.001	Significant

**Table 5 medicina-59-01580-t005:** Comparison between the variables for group A and group B at the initial vs. final assessment.

	PRE			POST		
Outcome Measure	T Value	*p* Value	Result (Significant/Not Significant)	T Value	*p* Value	Result (Significant/Not Significant)
AP	0.218	0.828	Not significant	−1.745	0.087	Significant
ML	−1.713	0.92	Not significant	−0.76	0.451	Not significant
MoT	−1.237	0.221	Not significant	0.85	0.399	Not significant
MJ	0.501	0.618	Not significant	3.711	<0.001	Significant
MC	−0.847	0.401	Not significant	0.646	0.521	Not significant
BR	−1.127	0.265	Not significant	−0.769	0.445	Not significant
DLB	0.363	0.718	Not significant	1.43	0.158	Not significant
DTB	0.921	0.361	Not significant	0.361	0.719	Not significant
DOB	−0.029	0.977	Not significant	0.293	0.005	Significant
DE	−0.867	0.389	Not significant	5.383	<0.001	Significant
TDR	0.145	0.885	Not significant	3.553	0.001	Significant
MB	0.706	0.483	Not significant	1.582	0.119	Not significant
MaT	−1.39	0.17	Not significant	2.736	0.008	Significant
MOB	−1.39	0.17	Not significant	0.889	0.378	Not significant
TMR	−1.179	0.244	Not significant	3.133	0.003	Significant

**Table 6 medicina-59-01580-t006:** The comparison of the results at Student *t*-tests at pre- and post-intervention moments for group A to group B.

	PRE			POST		
Outcome Measure	T Value	*p* Value	Result (Significant/Not Significant)	T Value	*p* Value	Result (Significant/Not Significant)
CVA	2.391	0.02	Significant	14.858	<0.001	Significant

## Data Availability

The datasets either used, analyzed, or both, during the current study are available from the corresponding authors on reasonable requests.

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
