# Peer review of "A Multidisciplinary Approach to Swallowing Rehabilitation in Patients with Forward Head Posture"

_medicina, 2023, doi:10.3390/medicina59091580_

Round 1

Reviewer 1 Report

Reviewer's Report

Title:

The title lacks specificity and does not clearly convey the main focus of the study. It should be more concise and provide a clear indication of the study's subject, such as "Effect of Combined Physiotherapy and Myofunctional Swallowing Rehabilitation on Forward Head Posture."

Abstract:

Negative point: The abstract is too brief and lacks important details about the study design, participants, and specific outcomes. It should include more comprehensive information, such as the sample size, inclusion/exclusion criteria, interventions used in both groups and specific findings in terms of improvement.

Introduction:

1. Lack of Background Information: The introduction starts with a brief definition of human posture but fails to provide a thorough background on forward head posture (FHP). The introduction should begin with a more comprehensive explanation of FHP, its prevalence, and its potential impact on swallowing function.

2. Missing Objective: The introduction does not clearly state the study's objective or research question. It should explicitly mention the main goal of the research, such as "The aim of this study is to investigate the effects of a combined physiotherapy and myofunctional swallowing rehabilitation program on forward head posture and its impact on swallowing function in healthy young individuals with atypical swallowing."

3. Limited Literature Review: The introduction briefly mentions the relationship between posture and deglutition but lacks an extensive review of relevant literature. A more in-depth review of previous studies on FHP, its impact on swallowing, and the effectiveness of rehabilitation programs would enhance the introduction's context and significance.

4. Inconsistent Citations: Some statements, such as "The modern lifestyle exposes people to dysfunctional postures," lack specific references to support the claims. Citations should be included to provide evidence for all factual information.

5. Organizational Structure: The introduction appears somewhat disjointed and could benefit from a more coherent flow of information. The logical progression of the information could be improved by organizing it into clear subsections that build upon each other.

Methods

1. Lack of control group: The study lacks a control group to compare the effects of the rehabilitation program with a group that did not receive any intervention. Without a control group, it is challenging to determine if the observed improvements are solely due to the rehabilitation program or other factors.

2. Small sample size: The study includes only 61 young patients, which may limit the generalizability of the findings. A larger sample size would increase the statistical power and improve the reliability of the results.

3. Selection bias: The inclusion criteria for the study were based on diagnoses made by different healthcare professionals (general dentists, orthodontists, general practitioners, neurologists, and physiotherapists). This may introduce selection bias as different practitioners may have different diagnostic criteria and thresholds.

4. Incomplete exclusion criteria: The exclusion criteria listed in the methods section do not cover all possible confounding factors. For instance, the study does not consider the effects of previous or ongoing treatments that may affect the study outcomes.

5. Lack of blinding: The study appears to lack blinding of the researchers and participants, which may introduce bias in the assessment of outcomes.

6. Lack of randomization: The assignment of patients into two groups (Group A and Group B) based on their choice and consent may introduce selection bias and confounding variables.

7. No mention of randomization method: If randomization was used, the manuscript does not specify the method used to achieve randomization, which raises concerns about the validity of the allocation process.

8. Limited details on assessment tools: The manuscript does not provide sufficient information about the reliability and validity of the assessment tools used (e.g., OMES-E, HATLFF, Payne technique). Without this information, it is challenging to evaluate the accuracy and consistency of the measurements.

9. Short follow-up period: The manuscript mentions that the rehabilitation program lasted for 20 weeks, but the follow-up period may not be sufficient to assess the long-term effectiveness of the intervention.

10. Lack of statistical details: The manuscript briefly mentions the use of statistical tests, but it lacks a detailed description of the statistical methods used for data analysis. This makes it difficult for readers to assess the appropriateness of the statistical analysis and the strength of the study's conclusions.

Results and Discussion

1. Lack of Statistical Details: The manuscript mentions significant differences in the outcome measures for both groups after the therapy, but it lacks specific statistical details, such as p-values or effect sizes. Without this information, it is challenging to assess the clinical significance of the observed differences.

2. Small Sample Size: The study's sample size is relatively small, with only 61 young patients aged 12-25 years. The small sample size may limit the generalizability of the results and reduce the study's statistical power. A larger and more diverse sample would be beneficial to strengthen the study's findings.

3. Lack of Control Group: The manuscript discusses two groups (group A with OMT and physiotherapy, and group B with only OMT), but it lacks a control group. Without a control group, it is difficult to determine whether the improvements observed are solely due to the interventions or could be attributed to other factors.

4. Insufficient Discussion of Limitations: While the manuscript briefly mentions some limitations, such as the small sample size and age range, it does not thoroughly discuss the potential impact of these limitations on the study's findings. A more comprehensive discussion of limitations and their implications would strengthen the paper's transparency and validity.

5. Ambiguous Terminology: The manuscript uses terms like "improved significantly" and "rehabilitated," but it does not clarify what these terms mean in the context of the study. The lack of clear definitions for the measured improvements makes it challenging to interpret the significance of the results.

6. Lack of Comparative Data: While the manuscript mentions that variables in group A were generally higher than in group B, it does not provide comparative data or effect sizes. Without this information, it is difficult to determine the magnitude of the differences between the two groups.

7. The Role of Multidisciplinary Approach: The manuscript suggests that a multidisciplinary approach, involving both myofunctional rehabilitation and physiotherapy, might yield better results. However, this conclusion is based on limited evidence from the current study, and it would be helpful to include references or previous research supporting this claim.

8. Causation vs. Correlation: The manuscript makes claims about the effect of the interventions on craniofacial growth and neuromuscular patterns, but it is essential to acknowledge that the study design does not establish causation. The observed correlations should be cautiously interpreted.

9. Lack of Comparison to Existing Literature: The manuscript claims that the study's results support previous findings, but it does not adequately compare and contrast its results with existing literature. A comprehensive discussion of how the current study aligns or differs from previous research would enhance the paper's contribution to the field.

10. Generalization Limitations: The manuscript discusses the findings in the context of patients with atypical swallowing and forward head posture. However, the limitations of the age range and the small sample size raise concerns about the generalizability of the results to a broader population.

In conclusion, the review report highlights several critical shortcomings in the results and discussion section of the manuscript. Addressing these issues, including providing more robust statistical details, clarifying ambiguous terminology, discussing limitations in-depth, and comparing findings to existing literature, would significantly improve the quality and impact of the study. Additionally, a larger and more diverse sample size, along with the inclusion of a control group, would enhance the generalizability of the results.

NIL

Author Response

Thank you for all the comments, they are very constructive and help us to add clarity to the study.

We completed the references and detailed the issues you highlighted in your comments. In addition, here are some of our points of view:

  • The title of the study expresses its focus on the swallowing rehabilitation itself; we wanted to emphasize that the combined therapy of the swallowing (not only on the oral structures, but the cranio-cervical muscles) leads to better results than the traditional, myofunctional approach, when dealing with FHP patients.
  • The control group is in fact group B, the one receiving only OMT for swallowing rehabilitation. Group A is the one that received combined therapy to verify our hypothesis that the FHP impacts the swallowing rehabilitation.
  • Although the patients were referred to the physiotherapist and speech therapist by different healthcare professionals, we consider that the initial assessment eliminated the possibility of bias since it was carried out at all patients with the same assessment tools and using the same diagnostic criteria.
  • We consider to extend the study on a larger sample of patients and also a larger age span.
  • We didn’t think about including these among the exclusion criteria because none of our patients benefitted or had benefitted of other treatments, but we should have mentioned this fact.
  • Although the assignment of the patients to one group or another was made upon their consent, they were only asked to choose one of the options, without being informed about the specific expectations (one approach expected to have better outcomes than the other). It is a voluntary participation and that is why we don’t consider the selection bias.
  • 20 weeks was the timeframe between the initial assessment and the final assessment. A 6 months follow-up is in progress, but for some patients, the 6 months have not yet passed since the end of the therapy

Yours sincerely, 

Judit Kurucz

Reviewer 2 Report

 Study Title:

A multidisciplinary approach to swallowing rehabilitation in patients with forward head posture

Reviewer’s Comments to Authors

Thanks for providing the opportunity to review such an interesting manuscript focused on a multidisciplinary approach to swallowing rehabilitation in patients with forward head posture. It is an interesting and novel study that will add value to the scientific world and Rehabilitation practices. The manuscript has been written well and concisely presented for easy reading. However, I would suggest at some points to make this manuscript the best version to be published if accepted.

Comments

Abstract

Line 14: The description of FHP requires more clarity. Suggest rephrasing: "The forward head posture (FHP) is characterized by increased extensions of upper cervical vertebrae and flexion of the lower cervical vertebrae and upper thoracic regions."

Line 15: There seems to be a subject-verb agreement error. Please correct to: "...spine negatively impacts suprahyoid..."

Line 18: For enhanced clarity and specificity, consider rewording: "The pattern of swallowing evolves gradually during childhood, solidifying by the age of four."

Line 19: The transition between normal and atypical swallowing isn't clear. Suggest: "If this developmental transition does not occur, the result is persistent infantile or 'atypical' swallowing."

Line 20: The term "abnormal biomechanical position" is somewhat vague. Consider refining to: "In individuals with FHP, muscles crucial to swallowing are biomechanically misaligned."

Line 22: The current phrasing could benefit from streamlining. Propose: "...lengthening of the suprahyoid muscles necessitates stronger contractions to achieve proper hyolaryngeal movement during swallowing."

Line 25: The objective of the research is broad. Refining the aim could help: "This study aims to assess the added benefits of physiotherapy to the traditional myofunctional swallowing rehabilitation for patients with FHP."

Line 27: The rationale's phrasing can be more direct. Suggest: "The underlying hypothesis is that without addressing FHP, swallowing rehabilitation remains challenged and potentially incomplete."

Line 28: The methods section would benefit from clarity in participant description. Suggest: "For the methods, 61 participants diagnosed with FHP and atypical swallowing..."

Line 30: There's redundancy in the mention of sessions. Consider: "...the other underwent both OMT and a weekly physiotherapy session for duration of 20 weeks."

Introduction

Line 52-70: Break up this section into smaller paragraphs for improved readability. The list of muscles affected (lines 60-68) is dense; consider simplifying or presenting in bullet format.

Line 71-82: The transition from FHP to the act of swallowing feels abrupt. Introduce a smoother transition sentence.

Line 83-101: For clarity, consider breaking this down into smaller paragraphs or using subheadings for each swallowing phase.

Line 112-122: Some sentences are long and intricate; consider breaking them up for better readability.

Line 132-141: To avoid redundancy, consider merging this section on abnormal swallowing with the earlier section discussing atypical swallowing patterns (line 76-81).

Methodology

Line 151-152: Mention the geographical location of the private hospital for context.

Line 158: Specify if parental consent was required for all minors or only those below a certain age.

Line 169: Analytic observational cohort study: Provide more details about why this type of study was chosen.

Line 174-176: Specify the reasons for these dropouts, if known.

Line178-179: Therapy sessions: Specify the sequence of the sessions. Did one type of session always precede the other?

Line 185: Assessment intervals: Indicate the time that passed between the initial and final assessments.

Line 219: Scoring: Provide a rationale for the scoring scale. Why a 4-point scale?

Line223: Healthy rest posture: Provide a citation or reference for this definition of "typical, healthy rest posture."

Line 230-231. Movement dysfunctions: Clarify how these dysfunctions were identified.

Lines 259-260: It's important to clarify the criteria for the scoring system. How are "deglutition" and "repetitions" being measured? If not consistently and accurately defined, it can lead to variability in scoring across assessors.

Line 321: The specifics of the camera setup (distance, height) are given, but it's essential to ensure that the focal length, angle, and settings are also standardized. These can significantly impact the results and repeatability of the photographs.

Line 404-407: This line seems to establish a cause and effect relationship ("FHP, with posterior cranial rotation... lead to an increased activity of the masticatory muscles and cranial extensors"). Ensure that this claim is backed by referenced studies or primary research data. If not, it should be rephrased as a hypothesis.

Lines 410-411: It mentions techniques and exercises that were partly the same for all patients. For technical accuracy, the commonalities and variations in techniques and exercises for different patients should be clearly listed or categorized

Statistical Analysis

Line 463-464: The term "t-Student" is not standard in English-language statistics literature. It's typically referred to as "paired t-test" or "Student's t-test." Consider revising for clarity.

Line 465-466: Similarly, replace "t-Student tests" with "independent t-tests." Additionally, the phrasing suggests you're comparing both before and after results between the two groups using two separate t-tests. Please clarify if this is the case.

Line 466-467:: Consider rephrasing for clarity: "The significance level was set at 0.05 unless otherwise specified." Also, specify whether the tests are one-tailed or two-tailed.

Results

Firstly, there is an inconsistency in the demographic details. The age range for women, as mentioned in the results, encompasses 12-26 years, which conflicts with the overall patient range of 12-25 years. Such discrepancies need to be rectified for clarity. Additionally, the reported percentage distribution for gender, summing to 108% (59% women and 49% men), requires correction to maintain credibility.

Regarding the abbreviation usage, terms like OMES-E, AP, ML, and others should either be fully expanded upon their first mention in the results section or be accompanied by a clear legend or footnote. Such clarification will be beneficial for readers who may be unfamiliar with these terms. Moreover, from a statistical presentation standpoint, P-values should ideally not be shown as an absolute "0". Extremely low P-values are conventionally presented as "<0.001" or a similar value, offering a more precise representation.

A separate table detailing patient characteristics would be immensely beneficial. This table could include details like age distribution, gender ratio, and other relevant demographic or baseline clinical information. Such a presentation would provide readers with a quick overview of the study population, aiding in the contextualization of the results. For Tables 3 and 4, which compare group A to group B, including descriptive statistics like means alongside the t-values could provide a more transparent interpretation.

In the tables presented, adding another column elucidating the variable's full name or its clinical relevance would offer a deeper understanding, especially when using abbreviations. This change would enhance clarity significantly. It's also vital to stress not just the statistical significance of the results but their clinical significance as well. Readers will benefit from understanding whether these changes have meaningful implications for the patient population.

Furthermore, the manuscript uses a P-value of 0.1 as a significant threshold, as observed in Table 3 for the variable AP. It's crucial to offer a rationale for this, as many studies traditionally employ a threshold of 0.05. If this threshold is domain-specific or backed by prior literature, it should be distinctly mentioned. Lastly, maintaining a consistent format across all tables, including uniform font size, column width, and spacing, will improve the manuscript's professional appearance and readability.

Discussion

Line 507-508: It would be beneficial to clarify which two groups you are referring to in the beginning of the discussion for those readers who might not remember the specific groupings from earlier sections. Consider introducing the two groups briefly once more.

Line 508-509: When listing the targeted structures (lips, tongue, jaw, etc.), ensure consistency. Either provide a comprehensive list at the start or reference it more generally without specifics.

Line 510-511: It's clear that group A underwent both OMT and physiotherapy, while group B only had OMT. However, emphasizing the inherent benefits or drawbacks of combining therapies in Group A might provide more context.

Line 512-515: The sentence structure here gets a bit convoluted, making it difficult for the reader to discern the differences between groups A and B for each variable. Consider breaking this down into simpler sentences or bullet points for clarity.

Line 517-518: You mention "most notable differences," but it's not clear what makes them "notable". Elaborating on this will give readers a better understanding of the significance.

Line 519-520: The relationship between impaired jaw movement and the rehabilitative effects seen in group A needs to be elucidated further. Were these improvements due to the combined therapies?

Line 522-525: The mention of "early goals in the OMT approach" is insightful. To enhance understanding, perhaps briefly remind the reader of what these early goals were in relation to the patient's struggles.

Line 526-528: The terms "external stability" and "internal stability" are introduced. Ensuring that these terms are either defined earlier in the paper or providing a brief definition here would be beneficial.

Line 529-533: The connection between resting posture and speech production is an interesting point. Consider expounding on the importance of a correct resting posture to emphasize its role in therapy.

Line 534-535: The significance of changing the breathing mode and its correlation to swallowing and chewing could use more emphasis or reference to earlier sections where this was discussed.

Line 538-539: The progression from an immature to mature pattern is crucial. Highlighting this transition with some supporting evidence or referencing earlier sections can help underscore its importance.

Line 542-544: The dramatic changes observed in group A, especially concerning the FHP and the mean angles, are significant. Perhaps provide a bit more context or a brief recap of these findings for readers who might not remember the specifics from the results section.

Line 546-547: When referencing the synergy between the suboccipital muscles, global chains, and other systems, ensure that all these components are explained earlier in the manuscript or provide a brief recap here

It needs to add limitations of the study.

Conclusion:

Rewrite your conclusion to show how your work advances the field, indicate the application of your work, and suggest future research.

Author Response

Thank you for your kind, interesting and constructive comments upon our study.

We made the suggested changes, rephrased, provide clarification and add more references where needed, in order to be clearer.

We chose the analytic observational study because we aimed to establish association between the exposure to a certain approach (single or combined) and the outcomes.

The 4-point scale is provided by OMES-E, a standardized and validated instrument for the assessment of the orofacial myofunctional disorders.

The movement dysfunctions were identified by observing the patient performing different movements (as specified in the study).

Yours sincerely,

Judit Mihaiu

Round 2

Reviewer 1 Report

I believe the authors have effectively responded to the reviewers' comments in the revised manuscript, so I have no additional remarks to add.

Nil

Reviewer 2 Report

After reviewing your manuscript, I am pleased to note significant improvements. I eagerly await to see your work in its published form.